# M2DF: Multi-grained Multi-curriculum Denoising Framework for Multimodal Aspect-based Sentiment Analysis

**Fei Zhao**[1*]    **Chunhui Li**[1*]    **Zhen Wu**[1†]    **Yawen Ouyang**[2]
**Jianbing Zhang**[1]    **Xinyu Dai**[1]

[1]National Key Laboratory for Novel Software Technology, Nanjing University, China
[2]Institute for AI Industry Research (AIR), Tsinghua University
{zhaof, lich}@smail.nju.edu.cn, ouyangyawen@air.tinghua.edu.cn
{wuz, zjb, daixinyu}@nju.edu.cn

## Abstract

Multimodal Aspect-based Sentiment Analysis (MABSA) is a fine-grained Sentiment Analysis task, which has attracted growing research interests recently. Existing work mainly utilizes image information to improve the performance of MABSA task. However, most of the studies overestimate the importance of images since there are many noisy images unrelated to the text in the dataset, which will have a negative impact on model learning. Although some work attempts to filter low-quality noisy images by setting thresholds, relying on thresholds will inevitably filter out a lot of useful image information. Therefore, in this work, we focus on whether the negative impact of noisy images can be reduced without filtering the data. To achieve this goal, we borrow the idea of Curriculum Learning and propose a Multi-grained Multi-curriculum Denoising Framework (M2DF), which can achieve denoising by adjusting the order of training data. Extensive experimental results show that our framework consistently outperforms state-of-the-art work on three sub-tasks of MABSA. Our code and datasets are available at https://github.com/grandchicken/M2DF.

## 1 Introduction

Multimodal Aspect-based Sentiment Analysis (MABSA) is a fine-grained Sentiment Analysis task (Liu, 2012; Pontiki et al., 2014), which has received extensive research attention in the past few years. MABSA has derived a number of sub-tasks, which all revolve around predicting several sentiment elements, i.e., aspect term(a), sentiment polarity(p), or their combinations. For example, given a pair of sentence and image, Multimodal Aspect Term Extraction (MATE) aims to extract all the aspect terms mentioned in a sentence (Wu et al., 2020a), Multimodal Aspect-Oriented Sentiment Classification (MASC) aims to identify the

---

* Equal contributions.
† Corresponding author.

**S:**    [Lady Gaga]$_{a_1}^{p_1:positive}$ out and about in [NYC]$_{a_2}^{p_2:neutral}$. ( May, 2 nd )

**I:** 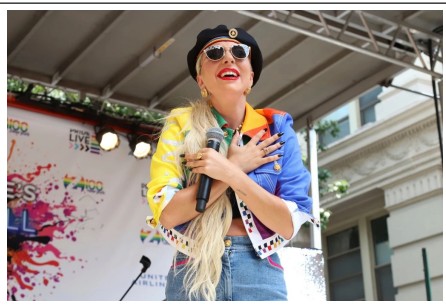

| Sub-tasks | Input | Output |
|---|---|---|
| Multimodal Aspect Term Extraction (MATE) | **S + I** | $a_1, a_2$ |
| Multimodal Aspect-Oriented Sentiment Classification (MASC) | **S + I** + $a_1$ | $p_1$ |
| | **S + I** + $a_2$ | $p_2$ |
| Joint Multimodal Aspect-Sentiment Analysis (JMASA) | **S + I** | $(a_1, p_1),$ $(a_2, p_2)$ |

Table 1: The definitions of all MABSA subtasks. Aspect terms and their corresponding sentiment polarities are highlighted in the sentence **S**.

sentiment polarity of a given aspect term in a sentence (Yu and Jiang, 2019), whereas the goal of Joint Multimodal Aspect-Sentiment Analysis (JMASA) is to extract all aspect terms and their corresponding sentiment polarities simultaneously (Ju et al., 2021). We illustrate the definitions of all the sub-tasks with a specific example in Table 1.

The assumption behind these sub-tasks is that the image information can help the text content identify the sentiment elements (Yu and Jiang, 2019; Ju et al., 2021), so how to make use of image information lies at the heart of the MABSA task. To this end, a lot of efforts have been devoted, roughly divided into three categories: (1) transforming the image into a global feature vector and utilizing attention mechanisms to sift through visual content pertinent to the text (Moon et al., 2018; Xu et al., 2019); (2) partitioning the image into multiple vi-

sual regions averagely, and subsequently establishing connections between text sequences and visual regions (Yu and Jiang, 2019; Yu et al., 2020; Ju et al., 2021); (3) retaining exclusively the regions containing visual objects in the image and enabling dynamic interactions with the text sequences (Wu et al., 2020a,b; Zhang et al., 2021; Yu et al., 2022; Ling et al., 2022; Yang et al., 2022).

Despite their promising results, most of the studies overestimate the importance of images since there are many noisy images unrelated to the text in the dataset, which will have a negative impact on model learning. In the MABSA task, Sun et al. (2020, 2021), Ju et al. (2021), Xu et al. (2022) and Yu et al. (2022) developed the cross-modal relation detection module that can filter low-quality noisy images by setting thresholds. However, relying on thresholds will inevitably filter out a lot of useful image information. Thus, in this work, we focus on another aspect: *can we reduce the negative impact of noisy images without filtering the data?*

To answer the above question, we borrow the idea of Curriculum Learning (CL) (Bengio et al., 2009), because CL can achieve denoising by adjusting the order of training data (Wang et al., 2022). In other words, CL no longer presents training data in a completely random order during training, but encourages training more time on clean data and less time on noisy data, which reveals the denoising efficacy of CL on noisy data. The key to applying CL is how to define clean/noisy examples and how to determine a proper denoising scheme. By analyzing the characteristics of the task, we tailor-design a **M**ulti-grained **M**ulti-curriculum **D**enoising **F**ramework (M2DF). M2DF first evaluates the degree of noisy images contained in each training instance from multiple granularities, and then the denoising curriculum gradually exposes the data containing noisy images to the model starting from clean data during training. In this way, M2DF theoretically assigns a larger learning weight to clean data (Gong et al., 2016), thereby achieving the effect of noise reduction.

Our contributions are summarized as follows: (1) To the best of our knowledge, we provide a new perspective to reduce the negative impact of noisy images in the MABSA task; (2) We propose a novel Multi-grained Multi-curriculum Denoising Framework, which is agnostic to the choice of base models; (3) We evaluate our denoising framework on some representative models, including the cur-

rent state-of-the-art. Extensive experimental results show that our denoising framework achieves competitive performance on three sub-tasks.

## 2 Related Work

In this section, we review the existing studies on Multimodal Aspect-based Sentiment Analysis (MABSA) and Curriculum Learning (CL).

**Multimodal Aspect-based Sentiment Analysis.** As an important sentiment analysis task, various neural networks have been proposed to deal with the three sub-tasks of MABSA, i.e., Multimodal Aspect Term Extraction (MATE), Multimodal Aspect-Oriented Sentiment Classification (MASC) and Joint Multimodal Aspect-Sentiment Analysis (JMASA). According to the different ways of utilizing image information, their work can be divided into three categories:

(1) Transforming the image into a global feature vector and utilizing well-designed attention mechanisms to sift through visual content pertinent to the text. Moon et al. (2018) used an attention mechanism to extract primary information from word embedding, character embedding, and global image vector. Xu et al. (2019) proposed a multi-interactive memory network to capture the multiple correlations in multimodal data.

(2) Partitioning the image into multiple visual regions averagely, and subsequently establishing connections between text sequences and visual regions. Yu and Jiang (2019) used the pre-trained model BERT (Devlin et al., 2019) and ResNet (He et al., 2016) to extract aspect and visual regions features, and devised an attention mechanism to obtain aspect-sensitive visual representations. Yu et al. (2020) proposed an entity span detection module to alleviate the visual bias. Sun et al. (2020), Sun et al. (2021) and Ju et al. (2021) introduced cross-modal relation detection module to decide whether the image feature is useful. Zhao et al. (2022a) proposed to utilize the external matching relations between different (text, image) pairs to improve the performance. Zhao et al. (2022b) developed a Knowledge-enhanced Framework to improve the visual attention capability.

(3) Retaining exclusively the regions containing visual objects in the image and enabling dynamic interactions with the text sequences. Wu et al. (2020a,b) and Zheng et al. (2021) employed Faster-RCNN (Ren et al., 2015) or Mask-RCNN (He et al., 2020) to extract object features, and then interact

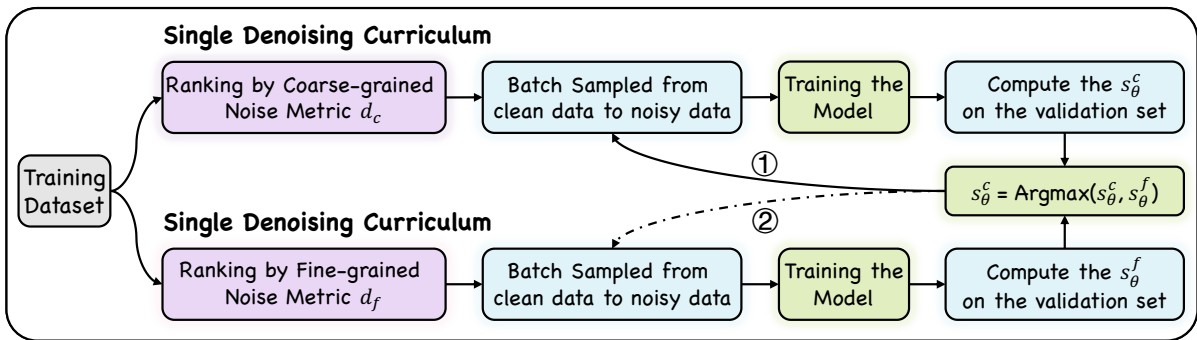

Figure 1: The overview of our M2DF framework. The solid line ① indicates the coarse-grained denoising curriculum is selected at the current training step, and the dashed line ② indicates it is not selected.

with textual features via a co-attention network. Zhang et al. (2021) utilized a multi-modal graph to capture various semantic relationships between words and visual objects in a (text, image) pair.

Despite their success, most studies overlooked the fact that there are many noisy images in the dataset that are unrelated to the text, which negatively impacts model learning. While some efforts (Sun et al., 2020, 2021; Yu et al., 2022; Ju et al., 2021) have been made to develop cross-modal relation detection modules to filter out low-quality noisy images, these approaches are threshold-dependent and inevitably filter out lots of useful image information. Therefore, in this work, we focus on whether the negative impact of noisy images can be reduced without data filtering.

**Curriculum Learning.** The concept of Curriculum Learning (CL) is initially introduced by (Bengio et al., 2009), and has since manifested its multifaceted advantages across a diverse array of machine learning tasks (Platanios et al., 2019; Wang et al., 2019; Liu et al., 2020; Xu et al., 2020; Lu and Zhang, 2021). One of the advantages of CL is denoising, which is achieved by training more time on clean data and less time on noisy data (Wang et al., 2022). In this work, we borrow the idea of CL to reduce the negative effect of noisy images. To the best of our knowledge, this is the first attempt to use CL for denoising in the MABSA task.

## 3 Methodology

### 3.1 Overview

As shown in Figure 1, we propose a Multi-grained Multi-curriculum Denoising Framework (M2DF) for the MABSA task, which consists of two modules: *Noise Metrics* and *Denoising Curriculums*. To be specific, when the image and text are unrelated, it is likely that the image is noise. In such cases, the similarity between the image and text is usually low, and the aspect terms in the text will generally not appear in the image. Based on this, we severally define a *coarse-grained (sentence-level) noise metric* and a *fine-grained (aspect-level) noise metric* to measure the degree of noisy images contained in each training instance.

Subsequently, we design a *single denoising curriculum* for each noise metric, which first ranks all training instances according to the size of noise metric, and then gradually exposes the training data containing noisy images to the model starting from clean data during training. In this manner, the model trains more time on the clean data and less time on the noisy data, so as to reduce the negative impact of noisy images. Given that the two noise metrics measure the noise level of the image from different granularities, a natural idea is to combine them together to produce better denoising effects. To this end, we extend the *single denoising curriculum* into *multiple denoising curriculum*.

### 3.2 Noise Metrics

In this section, we define the *Noise Metrics* used by M2DF from two granularities.

#### 3.2.1 Coarse-grained Noise Metric

Empirically, when the image and text are relevant, the similarity between them tends to be relatively high. Conversely, when the image and text are irrelevant, the similarity between them is generally low. Thus, it can be inferred that the lower the similarity between the image and the text, the more likely the image is noise.

Based on the above intuition, we measure the degree of noisy images contained in each training instance by calculating the similarity between the image and the text. Specifically, given a sentence-image pair $(S_i, I_i)$ in the training dataset $\mathcal{D}$, we first input each modality into different encoders of pre-trained model CLIP (Radford et al., 2021) to obtain the textual feature $H_S^i$ and visual feature $V_I^i$. Then the similarity is calculated as:

$$\text{sim}(H_S^i, V_I^i) = \cos(H_S^i, V_I^i), \qquad (1)$$

where $\cos(\cdot)$ is a cosine function. After that, considering the lower the similarity between the image and the text, the more likely the image is noise, we define a coarse-grained (sentence-level) noise metric as follows:

$$d_c = 1.0 - \frac{\text{sim}(H_S^i, V_I^i)}{\max_{(H_S^k, V_I^k) \in \mathcal{D}} \text{sim}(H_S^k, V_I^k)}, \quad (2)$$

where $d_c$ is normalized to [0.0, 1.0]. Here, a lower score indicates the pair $(S_i, I_i)$ is clean data.

### 3.2.2 Fine-grained Noise Metric

Except for sentence-level noise metric, we also measure the noise level from the perspective of aspect, because the goal of MABSA is to identify the aspect terms or predict the sentiment of aspect terms. Specifically, when the image and text are relevant, the aspect terms in the sentence will appear in the image with high probability. In contrast, when the image and text are irrelevant, the aspect terms in the sentence will generally not appear in the image. Thus, we can judge the relevance of the image and the text by calculating the similarity between the aspect terms in the sentence and the visual objects in the image. The lower the similarity is, the image is more likely to be noise.

Based on the above intuition, we measure the degree of noisy images by calculating the similarity between the aspect terms in the sentence and the visual objects in the image. However, except for the MASC task, the aspect terms of other subtasks are unknown. Considering that aspect terms are mostly noun phrases (Hu and Liu, 2004), we employ noun phrases extracted by the Stanford parser[1] as aspect terms within sentences for the MATE and JMASA tasks. Meanwhile, we use the object detection model Mask-RCNN to extract the visual objects in the image. The number of aspect terms

---

[1] https://nlp.stanford.edu/software/lex-parser.shtml

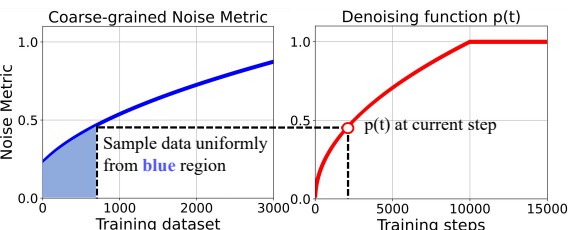

Figure 2: Illustration of the Single Denoising Curriculum for coarse-grained noise metric $d_c$. At each step, $p(t)$ is computed based on the current step $t$, and a batch of data is uniformly sampled from the training instances whose coarse-grained noise metric $d_c$ is lower than $p(t)$ (blue region in the example).

and visual objects extracted from the sentence $S_i$ and the image $I_i$ are as follows:

$$d_{aspect}(S_i) \triangleq N_{S_i}, \qquad (3)$$
$$d_{object}(I_i) \triangleq N_{I_i}, \qquad (4)$$

Then, we input aspect terms in sentence $S_i$ and visual objects in image $I_i$ into different encoders of pre-trained model CLIP to obtain a group of aspect features $\{H_a^{ij}\}_{j=1}^{N_{S_i}}$ and object features $\{V_o^{ij}\}_{j=1}^{N_{I_i}}$. After that, we average all these aspect features and object features to obtain the final aspect representation and object representation. The cross-modal similarity between them is as follows:

$$H_a^{S_i} = \frac{1}{N_{S_i}} \sum_{j=1}^{N_{S_i}} H_a^{ij}, \qquad (5)$$

$$V_o^{I_i} = \frac{1}{N_{I_i}} \sum_{j=1}^{N_{I_i}} V_o^{ij}, \qquad (6)$$

$$\text{sim}(H_a^{S_i}, V_o^{I_i}) = \cos(H_a^{S_i}, V_o^{I_i}), \qquad (7)$$

Finally, considering that the lower the similarity between the aspect terms and the visual objects, the more likely the image is noise, we define a fine-grained (aspect-level) noise metric as follows:

$$d_f = 1.0 - \frac{\text{sim}(H_a^{S_i}, V_o^{I_i})}{\max_{(H_a^{S_k}, V_o^{I_k}) \in \mathcal{D}} \text{sim}(H_a^{S_k}, V_o^{I_k})},$$
$$(8)$$

where $d_f$ is normalized to [0.0, 1.0]. Similarly, a lower score indicates the pair $(S_i, I_i)$ is clean data.

### 3.3 Denoising Curriculums

In this section, we first introduce the *single denoising curriculum* for each noise metric. Then we extend it into the *multiple denoising curriculum*.

**Algorithm 1:** Multiple Denoising Curriculum.

---

**Input:** The training set: $\mathcal{D}$; Coarse-grained Noise Metric: $d_c$; Fine-grained Noise Metric: $d_f$.
**Output:** A model with multiple denoising curriculum.

---

1   Compute two noise metrics $d_c$ and $d_f$ for each training sample in $\mathcal{D}$;
2   Sort $\mathcal{D}$ based on two noise metrics, resulting in $\mathcal{D}_c$ and $\mathcal{D}_f$;
   // Initialize $s_\theta^c$, $s_\theta^f$ for $\mathcal{D}_c$ and $\mathcal{D}_f$, respectively;
3   **for** $\mathcal{D}_i$ *in* $\mathcal{D}_c, \mathcal{D}_f$ **do**
4     Initialize the $p_i(0)$;
5     Initialize the training step $t_i = 0$;
6     **while** $t_i <= 1$ **do**
7       Uniformly sample one batch $B_i(t_i)$ from $\mathcal{D}_i$, such that $d_i \leq p_i(t_i)$;
8       Training the model with $B_i(t_i)$;
9       $t_i \leftarrow t_i + 1$;
10    **end**
11    Compute the ratio $s_\theta^i$ of two consecutive performances on a validation set;
12 **end**
   // Dynamically select the most suitable denoising curriculum;
13 **while** *not converged* **do**
14    $j = \underset{i \in \{c,f\}}{\arg\max}(s_\theta^i)$; // $j$ denotes $c$ or $f$
15    $t_j \leftarrow t_j + 1$;
16    Uniformly sample one batch $B_j(t_j)$ from $\mathcal{D}_j$, such that $d_j \leq p_j(t_j)$;
17    Training the model with $B_j(t_j)$;
18    Compute the ratio $s_\theta^j$ of two consecutive performances on a validation set;
19    Update $s_\theta^j$;
20 **end**

---

### 3.3.1   Single Denoising Curriculum

In training, we first sort each instance in the training dataset $\mathcal{D}$ based on a single coarse-grained noise metric $d_c$ or fine-grained noise metric $d_f$. Then inspired by (Platanios et al., 2019), we define a simple denoising function $p(t)$, which controls the pace of learning from clean data to noise instances. The specific function form is as follows:

$$p(t) = \begin{cases} \sqrt{t\dfrac{1 - p(0)^2}{T} + p(0)^2} & \text{if } t \leq T, \\ 1.0 & \text{otherwise.} \end{cases} \tag{9}$$

where $p(0)$ is a predefined initial value and is usually set to 0.01, $T$ is the duration of the denoising curriculum. At time step $t$, $p(t)$ means the upper limit of noise metric, and the model is only allowed to use the training instances whose noise metric $d_c$ or $d_f$ is lower than $p(t)$. In this way, the model no longer presents training data in completely random order during training, but trains more time on clean

data and less time on noisy data, which reveals the denoising efficacy of CL on noisy data (Wang et al., 2022). After $p(t)$ becomes 1.0, the single denoising curriculum is completed and the model can then freely access the entire dataset. In Figure 2, we give an illustration of the Single Denoising Curriculum for coarse-grained noise metric.

### 3.3.2   Multiple Denoising Curriculum

As mentioned above, each single denoising curriculum uses only one noise metric for denoising. Considering that the two noise metrics measure the noise level of the image from different granularities, a natural idea is to combine them together to produce better denoising effects. To this end, we present an extension to the single denoising curriculum: multiple denoising curriculum.

Intuitively, there are a few readily conceivable approaches, including directly merging two noise metrics into one (i.e., $d_c + d_f$), as well as randomly or sequentially selecting curricula to train the model (i.e., $d_c \rightarrow d_f \rightarrow d_c \rightarrow d_f$). However, a more impactful strategy lies in dynamically selecting the most appropriate denoising curricula at each training step. To achieve this goal, we use the ratio of two consecutive performances on a validation set to measure the effectiveness of each denoising curriculum as follows:

$$s_\theta = \frac{\chi^\theta}{\chi^{\theta_{prev}}}, \tag{10}$$

$$\chi^\theta = \frac{2 * Pre * Rec}{Pre + Rec}. \tag{11}$$

where $\chi^\theta$ is the F1-score computed at the current validation turn, $\chi^{\theta_{prev}}$ is computed at the previous validation turn, $Pre$ and $Rec$ denote Precision and Recall. A higher value of $s_\theta$ indicates greater progress achieved by the current denoising curriculum, which should lead to a larger sampling probability to achieve better denoising. Based on this, at each training step, we can dynamically select the most suitable denoising curriculum by comparing the values of $s_\theta$ between the two denoising curriculums. The details of multiple denoising curriculum are presented in Algorithm 1 and Figure 1.

## 4   Experiments

### 4.1   Experimental Settings

**Datasets.** To evaluate the effect of the M2DF Framework, we carry out experiments on two public multimodal datasets TWITTER-15 and

| | TWITTER-15 | | | | | | | TWITTER-17 | | | | | |
|---|---|---|---|---|---|---|---|---|---|---|---|---|---|
| | Pos | Neg | Neu | Total | AT | Words | AL | Pos | Neu | Neg | Total | AT | Words | AL |
| Train | 928 | 368 | 1883 | 3179 | 1.348 | 9023 | 16.72 | 1508 | 416 | 1638 | 3562 | 1.410 | 6027 | 16.21 |
| Dev | 303 | 149 | 670 | 1122 | 1.336 | 4238 | 16.74 | 515 | 144 | 517 | 1176 | 1.439 | 2922 | 16.37 |
| Test | 317 | 113 | 607 | 1037 | 1.354 | 3919 | 17.05 | 493 | 168 | 573 | 1234 | 1.450 | 3013 | 16.38 |

Table 2: The basic statistics of our two multimodal Twitter datasets. Pos: Positive, Neg: Negative, Neu: Neutral, AT: Avg. Targets, AL: Avg. Length

| Methods | TWITTER-15 | | | TWITTER-17 | | |
|---|---|---|---|---|---|---|
| | Pre | Rec | F1 | Pre | Rec | F1 |
| UMT-collapse (Yu et al., 2020) | 60.4 | 61.6 | 61.0 | 60.0 | 61.7 | 60.8 |
| + M2DF | 61.1±0.40 | 63.4±0.57 | 62.2±0.10 | 60.9±0.28 | 62.0±0.52 | 61.4±0.13 |
| OSCGA-collapse (Wu et al., 2020b) | 63.1 | 63.7 | 63.2 | 63.5 | 63.5 | 63.5 |
| + M2DF | 64.4±0.37 | 64.6±0.45 | 64.5±0.13 | 64.1±0.11 | 63.9±0.16 | 64.0±0.12 |
| RpBERT (Sun et al., 2021) | 49.3 | 46.9 | 48.0 | 57.0 | 55.4 | 56.2 |
| + M2DF | 49.3±0.20 | 49.0±0.25 | 49.2±0.15 | 56.9±0.34 | 56.5±0.38 | 56.7±0.22 |
| RDS♣ (Xu et al., 2022) | 60.8 | 61.7 | 61.2 | 61.8 | 62.9 | 62.3 |
| + M2DF | 61.2±0.12 | 63.0±0.35 | 62.1±0.15 | 62.4±0.16 | 63.6±0.12 | 63.0±0.08 |
| JML♣ (Ju et al., 2021) | 64.8 | 63.6 | 64.0 | 65.6 | 66.1 | 65.9 |
| + M2DF | 64.9±0.36 | 65.3±0.16 | 65.1±0.25 | 67.7±0.30 | 67.0±0.08 | 67.3±0.16 |
| VLP-MABSA♣ (Ling et al., 2022) | 64.1 | **68.6** | 66.3 | 65.8 | 67.9 | 66.9 |
| + M2DF | **67.0**±0.20 | 68.3±0.26 | **67.6**[†]±0.18 | **67.9**±0.10 | **68.8**±0.37 | **68.3**[†]±0.18 |

Table 3: Test results on the TWITTER-15 and TWITTER-17 datasets for JMASA task (%). For the baseline models, the results with ♣ are obtained by running the code released by the authors, and the other results without symbols are retrieved from (Ju et al., 2021) and (Ling et al., 2022). Best results are in bold. The marker [†] refers to significant test p-value $< 0.05$ when comparing with other multimodal methods.

TWITTER-17 from (Yu and Jiang, 2019), which include user tweets posted during 2014-2015 and 2016- 2017, respectively. General information for both datasets is presented in Table 2.

**Implementation Details.** During training, we train each model for a fixed 50 epochs, and then select the model with the best F1 score on the validation set. Finally, we evaluate its performance on the test set. We implement all the models with the PyTorch[2] framework, and run experiments on an RTX3090 GPU. More details about the hyper-parameters can be found in **Appendix** A.

**Evaluation Metrics and Significance Test.** As with the previous methods (Ju et al., 2021; Yu and Jiang, 2019), for MATE and JMASA tasks, we adopt Precision (Pre), Recall (Rec) and Micro-F1 (F1) as the evaluation metrics. For the MASC task, we use Accuracy (Acc) and Macro-F1 as evaluation metrics. Finally, we report the average performance and standard deviation over 5 runs with random initialization. Besides, the paired $t$-test is conducted to test the significance of different methods.

---

[2] https://pytorch.org/

### 4.2 Compared Methods

In this section, we introduce some representative baselines for each sub-task of MABSA: (1) **Baselines for Joint Multimodal Aspect-Sentiment Analysis (JMASA):** UMT-collapse (Yu et al., 2020), OSCGA-collapse (Wu et al., 2020b), Rp-BERT (Sun et al., 2021), RDS (Xu et al., 2022), JML (Ju et al., 2021), VLP-MABSA (Ling et al., 2022); (2) **Baselines for Multimodal Aspect Term Extraction (MATE):** UMT (Yu et al., 2020), OS-CGA (Wu et al., 2020b), JML (Ju et al., 2021), VLP-MABSA (Ling et al., 2022); (3) **Baselines for Multimodal Aspect Sentiment Classification (MASC):** TomBERT (Yu and Jiang, 2019), CapTr-BERT (Khan and Fu, 2021), JML (Ju et al., 2021), ITM (Yu et al., 2022), FITE (Yang et al., 2022), VLP-MABSA (Ling et al., 2022).

To verify the generalization of our framework, we apply M2DF to all baselines of three sub-tasks. It should be noted that RpBERT, RDS, JMT, and ITM use a cross-modal relation detection module to filter out noise images. In order to better compare their denoising effects with M2DF, we replace the cross-modal relation detection module in RpBERT, RDS, JMT, and ITM with our M2DF framework.

| Methods | TWITTER-15 | | | TWITTER-17 | | |
|---|---|---|---|---|---|---|
| | Pre | Rec | F1 | Pre | Rec | F1 |
| UMT (Yu et al., 2020) | 77.8 | 81.7 | 79.7 | 86.7 | 86.8 | 86.7 |
| + M2DF | 79.1±0.14 | 81.5±0.33 | 80.3±0.12 | 87.4±0.18 | 87.5±0.22 | 87.5±0.15 |
| OSCGA (Wu et al., 2020b) | 81.7 | 82.1 | 81.9 | 90.2 | 90.7 | 90.4 |
| + M2DF | 82.0±0.10 | 82.8±0.31 | 82.4±0.13 | 90.3±0.15 | 91.5±0.17 | 90.9±0.07 |
| JML♣ (Ju et al., 2021) | 82.9 | 81.2 | 82.0 | 90.2 | 90.9 | 90.5 |
| + M2DF | 84.0±0.26 | 82.3±0.12 | 83.1±0.14 | 91.1±0.11 | 90.9±0.18 | 91.0±0.12 |
| VLP-MABSA♣ (Ling et al., 2022) | 82.2 | **88.2** | 85.1 | 89.9 | 92.5 | 91.3 |
| + M2DF | **85.2**±0.24 | 87.4±0.20 | **86.3**$^\dagger$±0.15 | **91.5**±0.25 | **93.2**±0.23 | **92.4**$^\dagger$±0.14 |

Table 4: Test results on the TWITTER-15 and TWITTER-17 datasets for MATE task (%). For the baseline models, the results with ♣ are obtained by running the code released by the authors, and the other results without symbols are retrieved from (Ling et al., 2022). Best results are in bold.

| Methods | TWITTER-15 | | TWITTER-17 | |
|---|---|---|---|---|
| | Acc | F1 | Acc | F1 |
| TomBERT | 77.2 | 71.8 | 70.5 | 68.0 |
| + M2DF | 77.9±0.11 | 73.2±0.11 | 71.0±0.14 | 68.7±0.20 |
| CapTrBERT | 78.0 | 73.2 | 72.3 | 70.2 |
| + M2DF | 78.4±0.12 | 74.0±0.08 | 73.0±0.08 | 71.3±0.07 |
| FITE | 78.5 | 73.9 | 70.9 | 68.7 |
| + M2DF | 78.9±0.07 | 74.2±0.08 | 71.5±0.11 | 69.4±0.12 |
| ITM | 78.3 | 74.2 | 72.6 | 72.0 |
| + M2DF | **78.9**±0.05 | **75.0**±0.07 | 73.2±0.10 | **73.0**±0.08 |
| JML♣ | 78.1 | - | 72.7 | - |
| + M2DF | 78.8±0.15 | - | 74.0±0.12 | - |
| VLP-MABSA♣ | 77.2 | 72.9 | 73.2 | 71.4 |
| + M2DF | **78.9**$^\dagger$±0.15 | 74.8$^\dagger$±0.24 | **74.3**$^\dagger$±0.15 | **73.0**$^\dagger$±0.16 |

Table 5: Test results on two Twitter datasets for MASC task(%). For the baselines, the results with ♣ are obtained by running the code released by the authors, and the other results are retrieved from (Ling et al., 2022), (Yu et al., 2022) and (Yang et al., 2022). For a fair comparison, we follow the setting of (Ju et al., 2021) and (Ling et al., 2022), i.e., JML only evaluates the aspects correctly predicted on the MATE task while the other methods evaluate on all the golden aspects.

For other baselines, we directly integrate M2DF into them to obtain the final model.

## 5 Results and Analysis

### 5.1 Main Results

In this section, we analyze the results of different methods on three sub-tasks of MABSA.

**Results of JMASA.** The main experiment results of JMASA are shown in Table 3. Based on these results, we can make a couple of observations: (1) *VLP-MABSA* performs better than other multimodal baselines. A possible reason is that it develops a series of useful pre-training tasks; (2) Compared to the base model *UMT-collapse* and *OSCGA-collapse*, *UMT-collapse + M2DF* and *OSCGA-collapse + M2DF* achieve better results

on both datasets. This outcome reveals the effective denoising effect of our framework; (3) It is worth mentioning that *RpBERT + M2DF* and *RDS + M2DF* obtain good improvements in Micro-F1 over *RpBERT* and *RDS*. Similarly, *JMT + M2DF* achieves a boost of 1.1% and 1.4% in Micro-F1 compared to *JMT*. These observations indicate that the denoising effect of M2DF is better than that of the cross-modal relation detection module; (4) The performance boost of *VLP-MABSA + M2DF* over *VLP-MABSA* by 1.3% and 1.4%. These results affirm the robustness of the M2DF, showcasing its ability to consistently generate substantial enhancements even when applied to the current state-of-the-art model.

**Results of MATE and MASC.** Table 4 and Table 5 show the main experiment results of MATE and MASC. As we can see, incorporating M2DF into the baselines of MATE and MATC tasks can obtain competitive performance. To be specific, for the MATE task, *VLP-MABSA + M2DF* obtain 1.2% and 1.1% improvements in terms of the F1-score. For the MASC task, *VLP-MABSA + M2DF* can improve the F1-score by up to 1.9% and 1.6%. These results indicate that M2DF is applicable to three subtasks simultaneously, showcasing its remarkable capacity for generalization.

### 5.2 Ablation Study

Without loss of generality, we choose *VLP-MABSA + M2DF* model for the ablation study to investigate the effects of different modules in M2DF.

**Effects of the Noise Metrics.** As we mentioned in Section 3.3.1, we develop a single denoising curriculum for each noise metric. Here, we discuss the contribution of each noise metric in our framework. From Table 6, we can observe the fol-

| Settings | Coarse-grained Noise Metric | Fine-grained Noise Metric | Denoising Curriculum | TWITTER-15 | | | TWITTER-17 | | |
|---|---|---|---|---|---|---|---|---|---|
| | | | | MATE | MASC | JMASA | MATE | MASC | JMASA |
| VLP-MABSA | ✘ | ✘ | ✘ | 85.1 | 72.9 | 66.3 | 91.3 | 71.4 | 66.9 |
| (a) | ✔ | ✘ | Single | 85.8 | 73.6 | 66.8 | 91.6 | 72.1 | 67.4 |
| (b) | ✘ | ✔ | Single | 85.9 | 73.9 | 67.0 | 91.6 | 72.3 | 67.5 |
| (c) | ✔ | ✔ | Multiple | 86.3 | 74.8 | 67.6 | 92.4 | 73.0 | 68.3 |
| (d) | ✔ | ✔ | Merge | 85.8 | 73.8 | 66.9 | 91.7 | 72.1 | 67.9 |
| (e) | ✔ | ✔ | Randomly | 86.1 | 74.2 | 67.3 | 91.9 | 72.5 | 68.0 |
| (f) | ✔ | ✔ | Sequentially | 86.2 | 74.5 | 67.3 | 92.1 | 72.6 | 68.2 |

Table 6: Ablation study of our denoising framework(%). We use the current state-of-the-art model VLP-MABSA (Ling et al., 2022) as the base model to conduct the analysis. The setting (c) denotes our full proposed approach. We evaluate three tasks JMASA, MATE, and MASC in terms of F1.

| Methods | Level-1 | | | Level-2 | | | Level-3 | | |
|---|---|---|---|---|---|---|---|---|---|
| | MATE | MASC | JMASA | MATE | MASC | JMASA | MATE | MASC | JMASA |
| VLP-MABSA | 92.7 | 73.4 | 67.2 | 91.6 | 73.0 | 69.8 | 89.6 | 68.0 | 63.4 |
| VLP-MABSA + M2DF | 92.9 | 73.6 | 67.8 | 92.1 | 73.8 | 71.2 | 91.6 | 71.3 | 65.6 |
| Δ | +0.2 | +0.2 | +0.6 | +0.5 | +0.8 | +1.4 | +2.0 | +3.3 | +2.2 |

Table 7: Model performance on the divided TWITTER-17 test sets with different noise levels (%). We evaluate three tasks JMASA, MATE, and MASC in terms of F1. Δ represents the difference between the performance of VLP-MABSA and VLP-MABSA + M2DF at different noise level.

lowing: (1) Settings (a-b) reveal that every noise metric can boost the performance of base model *VLP-MABSA*, which validates the rationality of our designed noise metrics; (2) The fine-grained noise metric achieves better performance than the coarse-grained noise metric in most tasks, which indicates that fine-grained noise metric is more accurate in measuring whether noisy images are included in each training instance; (3) When incorporate two noise metric together, the performance is further improved (see setting c), which demonstrates that two noise metrics improve the performance of the MABSA task from different perspectives.

**Effects of the Denoising Curriculums.** As we mentioned in Section 3.3.2, three easily thought of strategies to implement the multiple denoising curriculum are Merge, Randomly, and Sequentially. Table 6 (settings d-f) shows the results of the three strategies. As we can see, (1) All three strategies achieve stable improvements on the MABSA tasks, which demonstrates the effectiveness and robustness of our M2DF framework; (2) The results of the Merge strategy are inferior to Randomly and Sequentially. A possible reason is that directly merging two noise metrics is unable to give full play to their respective advantages; (3) All of them perform worse than our final approach (setting c), which verifies the superiority of the dynamic learning strategy at each training step.

**Performance on Different Subsets.** To gain a deeper insight into the role of M2DF, we took an additional step by partitioning the TWITTER-17 test set into three subsets of equal size. This division was based on the values of a coarse-grained noise metric $d_c$, spanning from low to high. By employing this approach, we aimed to examine the impact of M2DF across various levels of noise present in the dataset. The results are shown in Table 7. We can observe that compared with *VLP-MABSA*, *VLP-MABSA + M2DF* achieves more performance improvement on the subset with the higher noise level. This suggests that M2DF can indeed mitigate the negative impact of noisy images.

### 5.3 Training Time

Curriculum Learning can effectively steer the learning process away from poor local optima, making it converge faster (Bengio et al., 2009; Platanios et al., 2019). As a result, the training time for the denoising phase is reduced compared to regular training. In **Appendix** C, we display the training times for VLP-MABSA and VLP-MABSA+M2DF on the JMASA task.

### 5.4 Case Study

As we mentioned in section 5.2, M2DF performs well when the multimodal instance contains noisy images unrelated to the text, here we conduct the case study to better understand the advantages of

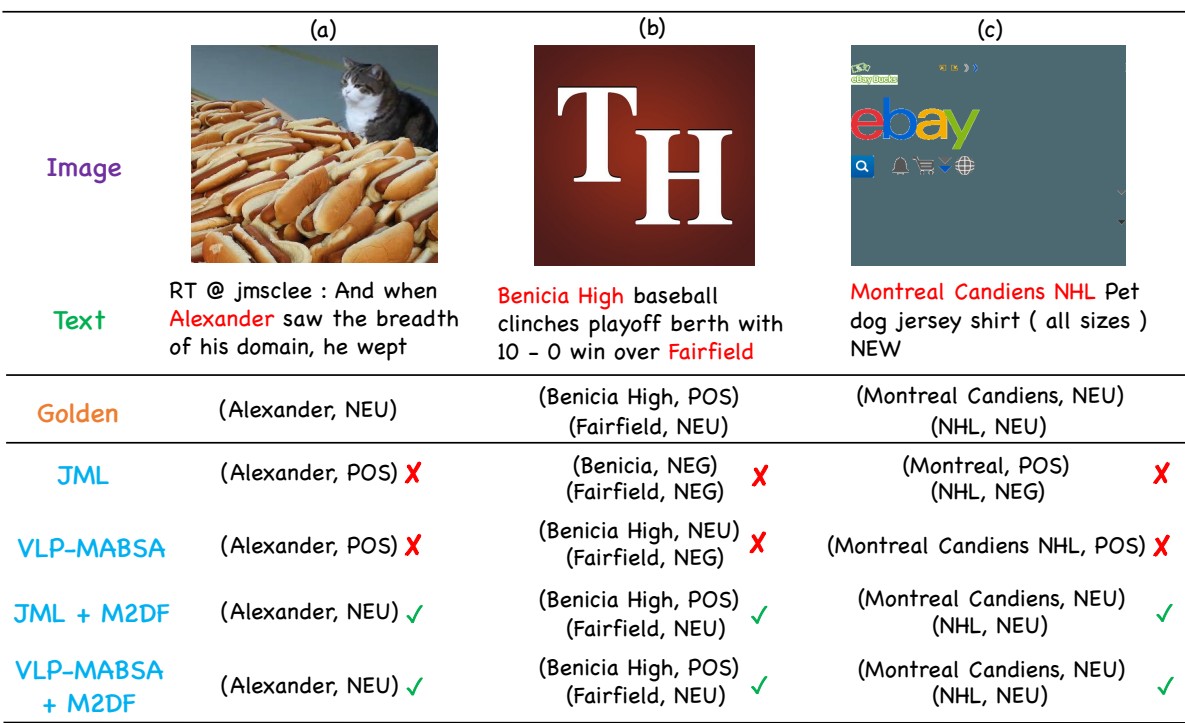

| | (a) | (b) | (c) |
|---|---|---|---|
| Image | | | |
| Text | RT @ jmsclee : And when Alexander saw the breadth of his domain, he wept | Benicia High baseball clinches playoff berth with 10 - 0 win over Fairfield | Montreal Candiens NHL Pet dog jersey shirt ( all sizes ) NEW |
| Golden | (Alexander, NEU) | (Benicia High, POS) (Fairfield, NEU) | (Montreal Candiens, NEU) (NHL, NEU) |
| JML | (Alexander, POS) ✗ | (Benicia, NEG) (Fairfield, NEG) ✗ | (Montreal, POS) (NHL, NEG) ✗ |
| VLP-MABSA | (Alexander, POS) ✗ | (Benicia High, NEU) (Fairfield, NEG) ✗ | (Montreal Candiens NHL, POS) ✗ |
| JML + M2DF | (Alexander, NEU) ✓ | (Benicia High, POS) (Fairfield, NEU) ✓ | (Montreal Candiens, NEU) (NHL, NEU) ✓ |
| VLP-MABSA + M2DF | (Alexander, NEU) ✓ | (Benicia High, POS) (Fairfield, NEU) ✓ | (Montreal Candiens, NEU) (NHL, NEU) ✓ |

Figure 3: Three examples of the predictions by JML, VLP-MABSA, JML + M2DF, and VLP-MABSA + M2DF. POS, NEU, and NEG are short for Positive, Neutral, and Negative, respectively.

M2DF. Figure 3 presents the predict results of three noise examples on the JMASA task by *JML + M2DF* and *VLP-MABSA + M2DF*, as well as two representative baselines *JML* and *VLP-MABSA*. It is clear that both *JML* and *VLP-MABSA* give the incorrect prediction when faced with noisy images. The reason behind this may be that: (1) although *JML* develops a cross-modal relation detection module to filter out noisy images, this module relies on thresholds, so using it to filter noisy images is not reliable enough; (2) *VLP-MABSA* treats the image equally and ignores the negative effects of noisy images. In contrast, we observe that *JML + M2DF* and *VLP-MABSA + M2DF* can obtain all correct aspect terms and aspect-dependent sentiment on three noise samples, suggesting that our framework M2DF can achieve better denoising.

## 6 Conclusion

In this paper, we propose a novel Multi-grained Multi-curriculum Denoising Framework (M2DF) for the MABSA task. Specifically, we first define a coarse-grained noise metric and a fine-grained noise metric to measure the degree of noisy images contained in each training instance. Then, in order to reduce the negative impact of noisy images, we design a single denoising curriculum and a multi-

ple denoising curriculum. Results from numerous experiments indicate that our denoising framework achieves better performance than other state-of-the-art methods. Further analysis also validates the superiority of our denoising framework.

## Limitations

The current study has two limitations that warrant further attention. First, while we have developed two noise metrics to evaluate the degree of noisy images contained in the dataset, there may be other methods to achieve this goal, which requires us to do further research and exploration. Second, we have not examined other CL training schedules, such as self-paced learning (Kumar et al., 2010), which may be worth considering in the future. We view our study as a starting point for future research, with the goal of propelling the field forward and leading to even better model performance.

## Acknowledgements

We would like to thank the anonymous reviewers for their insightful comments. Fei Zhao would like to thank Siyu Long for his constructive suggestions. This work is supported by the National Natural Science Foundation of China (No. 62206126, 61936012 and 61976114).

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

## A    Implementation Details

As we mentioned in section 4.2, we chose some representative models as the foundations of our work for three sub-tasks. Therefore, when applying our framework, we keep the parameters the same as the baseline methods in Table 8. We implement all the models with the PyTorch framework, and run experiments on an RTX3090 GPU.

## B    Additional Analysis

**The Image Information is Useful**    Except for multimodal baselines, we also compare the M2DF with pure textual ABSA models, i.e., SPAN (Hu et al., 2019), D-GCN (Chen et al., 2020) and BART (Lewis et al., 2020), the results of the JMASA task are shown in Table 9. For text-based methods, it is clear that BART consistently outperforms all the baselines, we attribute this to the fact that the encoder-decoder structure of BART can grasp the overall meaning of the whole sentence in advance, which is important for predicting label sequences. VLP-MABSA is built upon the

| Methods | Backbone | Learning Rate |
|---|---|---|
| UMT-collapse | BERT+ResNet | 5e-5 |
| UMT | BERT+ResNet | 5e-5 |
| OSCGA-collapse | Glove+Mask RCNN | 0.008 |
| OSCGA | Glove+Mask RCNN | 0.008 |
| RpBERT | BERT+ResNet | 1e-6 |
| TomBERT | BERT+ResNet | 5e-5 |
| CapTriBERT | BERT+ResNet | 5e-5 |
| FITE | BERT+ResNet | 5e-5 |
| JML | BERT+ResNet | 2e-5 |
| VLP-MABSA | BART+Faster R-CNN | 5e-5 |

Table 8: Hyperparameters of baseline methods.

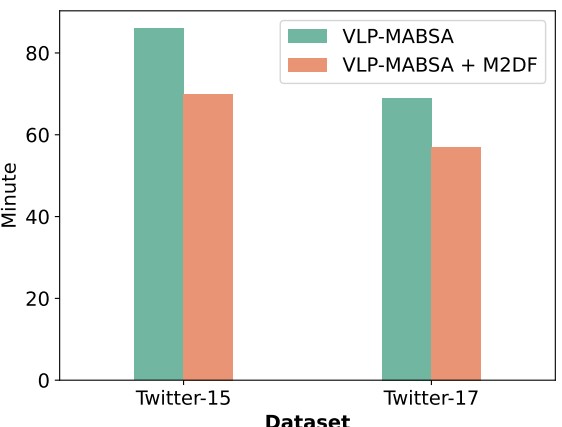

Figure 4: Training times for VLP-MABSA and VLP-MABSA+M2DF on the JMASA task.

foundation of BART, and the multimodal methods VLP-MABSA and VLP-MABSA + M2DF both outperform BART, which indicates that image information is useful for the MABSA task. This aligns with the conclusions of previous methods.

**Effect of Different Multimodal Encoder and Image Encoder**    We utilize CLIP for multimodal feature extraction, due to its training on a vast collection of text-image pairs. This allows it to excel in capturing multimodal semantics better. Besides, we also try using ViLBERT (Lu et al., 2019) to extract multimodal features. For object detection, we try using Faster-RCNN to replace Mask-RCNN. The results of these two methods on the JMASA task are shown in Table 10. We can observe that the F1-score of VLP-MABSA + M2DF (ViLBERT) is lower than that of VLP-MABSA + M2DF (Ours). This is because CLIP benefits from a larger-scale and more diverse training dataset, which enables the model to capture multimodal features more accurately. Thus, CLIP is a better version to extract multimodal features. Moreover, the F1-score of VLP-MABSA + M2DF (Faster-RCNN) is slightly

| Methods | TWITTER-15 | | | TWITTER-17 | | |
|---|---|---|---|---|---|---|
| | Pre | Rec | F1 | Pre | Rec | F1 |
| SPAN (Hu et al., 2019) | 53.7 | 53.9 | 53.8 | 59.6 | 61.7 | 60.6 |
| D-GCN (Chen et al., 2020) | 58.3 | 58.8 | 58.5 | 64.2 | 64.1 | 64.1 |
| BART (Lewis et al., 2020) | 62.9 | 65.0 | 63.9 | 65.2 | 65.6 | 65.4 |
| VLP-MABSA (Ling et al., 2022) | 64.1 | 68.6 | 66.3 | 65.8 | 67.9 | 66.9 |
| VLP-MABSA + M2DF | 67.0±0.20 | 68.3±0.26 | 67.6±0.18 | 67.9±0.10 | 68.8±0.37 | 68.3±0.18 |

Table 9: Test results on the `TWITTER-15` and `TWITTER-17` datasets for the JMASA task (%). The results of text-based methods (i.e., SPAN, D-GCN, and BART) are retrieved from (Ling et al., 2022).

| Methods | TWITTER-15 | | | TWITTER-17 | | |
|---|---|---|---|---|---|---|
| | Pre | Rec | F1 | Pre | Rec | F1 |
| VLP-MABSA + M2DF (Ours) | 67.0±0.20 | 68.3±0.26 | 67.6±0.18 | 67.9±0.10 | 68.8±0.37 | 68.3±0.18 |
| VLP-MABSA + M2DF (ViLBERT) | 66.6±0.09 | 67.9±0.25 | 67.2±0.11 | 67.3±0.29 | 68.2±0.13 | 67.7±0.17 |
| VLP-MABSA + M2DF (Faster-RCNN) | 66.9±0.40 | 68.1±0.18 | 67.5±0.11 | 67.7±0.18 | 68.6±0.18 | 68.1±0.09 |

Table 10: Effect of different image encoders (%).

lower than that of VLP-MABSA + M2DF (Ours), which indicates that Mask-RCNN is a better version for object detection in the MABSA task. Certainly, Faster-RCNN is also a good choice.

## C Training Time

Figure 4 lists the training times for VLP-MABSA and VLP-MABSA+M2DF on the JMASA task.

## D Error Analysis

We randomly sample 100 error cases of *VLP-MABSA + M2DF* in the JMASA task, and then divide them into three error categories. Figure 5 shows the proportions and some representative examples for each category. The top category is annotation bias. As shown in Figure 5(a), the sentiment of aspect term "Orange Carpet" is annotated as "NEU" (Neutral), but it actually expresses the sentiment of "POS" (Positive). This type of error presents a significant challenge for our model to provide accurate predictions. The second category is lacking background knowledge. In Figure 5(b), the aspect term "Orange is the New Black" refers to a well-known comedy, the model typically can only extract part of the aspect term without the help of background knowledge. The third category is insufficient information. As shown in Figure 5(c), both the sentence and the image content are quite simple, which is unable to provide sufficient information for our model to accurately identify aspect terms and their corresponding sentiment polarities.

There is a need for the development of more advanced natural language processing techniques to address the above problems.

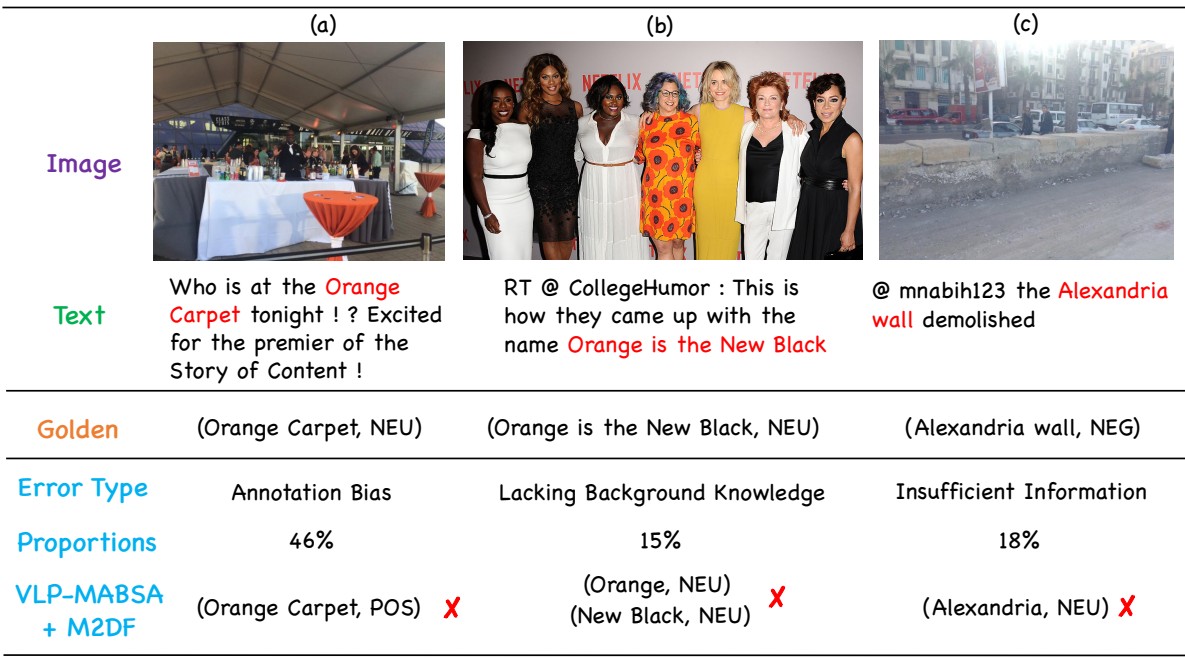

|  | (a) | (b) | (c) |
|---|---|---|---|
| Image | | | |
| Text | Who is at the Orange Carpet tonight ! ? Excited for the premier of the Story of Content ! | RT @ CollegeHumor : This is how they came up with the name Orange is the New Black | @ mnabih123 the Alexandria wall demolished |
| Golden | (Orange Carpet, NEU) | (Orange is the New Black, NEU) | (Alexandria wall, NEG) |
| Error Type | Annotation Bias | Lacking Background Knowledge | Insufficient Information |
| Proportions | 46% | 15% | 18% |
| VLP-MABSA + M2DF | (Orange Carpet, POS) ✗ | (Orange, NEU) (New Black, NEU) ✗ | (Alexandria, NEU) ✗ |

Figure 5: Three typical errors of VLP-MABSA + M2DF. POS: Positive, NEU: Neutral, NEG: Negative.