# OpenReview forum: "M2DF: Multi-grained Multi-curriculum Denoising Framework for Multimodal Aspect-based Sentiment Analysis"
_EMNLP/2023/Conference — EMNLP 2023 Main_

### Official Review · Reviewer_Szeq · 2023-08-01

**Typos Grammar Style And Presentation Improvements:** 1. In section 3.1 paragraph1, the deg…
**Soundness:** 3

**Excitement:**

3: Ambivalent: It has merits (e.g., it reports state-of-the-art results, the idea is nice), but there are key weaknesses (e.g., it describes incremental work), and it can significantly benefit from another round of revision. However, I won't object to accepting it if my co-reviewers champion it.

**Paper Topic And Main Contributions:**

This paper targets the Multimodal Aspect-based Sentiment Analysis task. One of the problems with the current studies of the task is that the importance of image information is overrated and noisy images often bring a negative impact on the sentiment analysis result. Therefore, the authors propose a novel Multi-grained Multi-curriculum Denoising Framework inspired by Curriculum Learning to denoise training data by reordering data samples to train more times on clean data.
The main contributions of the paper are:
1. Proposing a new perspective on MASBA noise reduction.
2. Proposing a novel Multimodal Aspect-base Sentiment Analysis.
3. Conducting extensive experiments to evaluate the model.


**Questions For The Authors:**

Question A: The proposed framework is composed of several neural networks, such as CLIP for multimodal feature extraction and Mask-RCNN for object detection. Is the current framework the best version? Have you tried any other neural models for these purposes?
Question B: As this paper argues for the overestimated importance of image information, you should also compare the framework with pure textual ABSA models to prove that the image information is still useful.


**Reasons To Accept:**

This paper introduces a novel idea to denoise image data through reordering training samples. The noise of the image is measured comprehensively, from both the sentence-level and aspect-level. Experiments prove the effectiveness of the proposed framework.

**Reasons To Reject:**

1. For the JMASA task, the improvement of adding the M2DF framework is very little, and VLP-MABSA + M2DF performs even worse than the original VLP-MABSA model in terms of recall. What are the potential drawbacks of the proposed framework that lead to this result?

**Reproducibility:**

4: Could mostly reproduce the results, but there may be some variation because of sample variance or minor variations in their interpretation of the protocol or method.

**Reviewer Confidence:**

5: Positive that my evaluation is correct. I read the paper very carefully and I am very familiar with related work.

---

> ### Author Rebuttal · Authors · 2023-08-28
>
> Thanks for your appreciation of our new perspective, novel idea, and extensive experiments. We address your concerns as follows:
>
> __Q1: The proposed framework is composed of several neural networks, such as CLIP for multimodal feature extraction and Mask-RCNN for object detection. Is the current framework the best version? Have you tried any other neural models for these purposes?__
>
> __A1:__ We utilized CLIP for multimodal feature extraction, due to its training on a vast collection of text-image pairs. This allows it to excel in capturing multimodal semantics better. Besides, we also tried using ViLBERT[5] to extract multimodal features. For object detection, we tried using Faster-RCNN to replace Mask-RCNN. The results of these two methods on the JMASA task are as follows:
>
> Methods | TWITTER-15(Pre, Rec, F1) | TWITTER-17(Pre, Rec, F1) |
> | :--- | :----: | :----: |
> VLP-MABSA + M2DF(Ours) | 67.0 68.3 67.6 | 67.9 68.8 68.3 |
> VLP-MABSA + M2DF(ViLBERT) | 66.6 67.9 67.2 | 67.3 68.2 67.7 |
> VLP-MABSA + M2DF(Faster-RCNN) | 66.9 68.1 67.5 | 67.7 68.6 68.1 |
>
> We can observe that the F1-score of VLP-MABSA + M2DF(ViLBERT) is lower than that of VLP-MABSA + M2DF(Ours). This is because CLIP benefits from a larger-scale and more diverse training dataset, which enables the model to capture multimodal features more accurately. Thus, CLIP is a better version to extract multimodal features. Moreover, the F1-score of VLP-MABSA + M2DF(Faster-RCNN) is slightly lower than that of VLP-MABSA + M2DF(Ours), which indicates that Mask-RCNN is a better version for object detection in the MABSA task. Certainly, Faster-RCNN is also a good choice.
>
> [5] ViLBERT: Pretraining Task-Agnostic Visiolinguistic Representations for Vision-and-Language Tasks. NeurIPS 2019.
>
>
> __Q2: As this paper argues for the overestimated importance of image information, you should also compare the framework with pure textual ABSA models to prove that the image information is still useful.__
>
> __A2:__ Thanks for your constructive suggestion. Due to space constraints, we didn't report the previous text-based ABSA methods in the main results. In the following, we retrieve the results of text-based methods (i.e., SPAN, D-GCN, and BART) from [6] for the JMASA task.
>
> Methods | TWITTER-15(Pre, Rec, F1) | TWITTER-17(Pre, Rec, F1) |
> | :--- | :----: | :----: |
> SPAN | 53.7 53.9 53.8 | 59.6 61.7 60.6 |
> D-GCN | 58.3 58.8 58.5 | 64.2 64.1 64.1 |
> BART | 62.9 65.0 63.9 | 65.2 65.6 65.4 |
> VLP-MABSA | 64.1 68.6 66.3 | 65.8 67.9 66.9 |
> VLP-MABSA + M2DF | 67.0 68.3 67.6 | 67.9 68.8 68.3 |
>
> It is clear that multimodal methods VLP-MABSA and VLP-MABSA + M2DF both outperform all the text-based methods, which indicates that the image information is useful for MABSA. We will add these results to our revision. Thanks for your suggestion again.
>
> [6] Vision-Language Pre-Training for Multimodal Aspect-Based Sentiment Analysis. ACL 2022.
>
>
> __Q3: For the JMASA task, VLP-MABSA + M2DF performs worse than the original VLP-MABSA model in terms of recall. What are the potential drawbacks of the proposed framework that lead to this result?__
>
> __A3:__ Thanks for your valuable comments. Compared to VLP-MABSA, VLP-MABSA + M2DF achieves a significant improvement in the precision, but at the same time, the recall has a certain degree of sacrifice, that is, the recall is slightly decreased or the improvement is not as obvious as the precision. This phenomenon might stem from M2DF's heightened stringency in prediction generation within the sequence-to-sequence framework of VLP-MABSA. M2DF tends to generate accurate predictions with high confidence, resulting in a substantial enhancement in precision by reducing the number of erroneous predictions. However, the heightened stringency in prediction generation by M2DF may inadvertently lead to the omission of some true instances, thereby resulting in a certain degree of sacrifice in recall. But overall, the F1-score achieves competitive results.
>
>
> __Q4: About typos.__
>
> __A4:__ Thanks for your careful review. We will fix these typos in the revision.

---

### Official Review · Reviewer_sBBK · 2023-08-04

**Soundness:** 3

**Excitement:**

3: Ambivalent: It has merits (e.g., it reports state-of-the-art results, the idea is nice), but there are key weaknesses (e.g., it describes incremental work), and it can significantly benefit from another round of revision. However, I won't object to accepting it if my co-reviewers champion it.

**Paper Topic And Main Contributions:**

This paper introduces a Multi-grained Multi-curriculum Denoising Framework (M2DF) to tackle the problem of Multimodal Aspect-based Sentiment Analysis (MABSA). They focus on whether the negative impact of noisy images can be reduced without filtering the data. Extensive experimental results show that the proposed framework consistently outperforms state-of-the-art work on three sub-tasks of MABSA.

**Questions For The Authors:**

1. Did authors try other noise metrics to evaluate the degree of noisy images? How is the performance like?

**Reasons To Accept:**

1. This work is supposed to be the first attempt to use curriculum learning for denoising in the MABSA task.
2. Several lines of recent work are included in the baselines and the results show the proposed method can improve the performance.

**Reasons To Reject:**

1. This paper is not well-organized. Implementation details can be put into the appendix to leave some space for the analysis part. Case study and error analysis are better to be put into the main content.
2. The motivation is not clear. Authors should explain more on why we need curriculum learning in this work. Can we use other machine learning algorithms? Also, this paper only show curriculum learning schedule and readers might wonder whether other training schedules such as self-paced learning work.
3. More training schedules should be involved as baselines in the experiments to ensure fair comparisons, since this work is a new training schedule.

**Reproducibility:**

3: Could reproduce the results with some difficulty. The settings of parameters are underspecified or subjectively determined; the training/evaluation data are not widely available.

**Reviewer Confidence:**

4: Quite sure. I tried to check the important points carefully. It's unlikely, though conceivable, that I missed something that should affect my ratings.

---

> ### Author Rebuttal · Authors · 2023-08-28
>
> Thanks for your appreciation of our first attempt to use curriculum learning for denoising in the MABSA task. About the mentioned questions, our answers are as follows:
>
> __Q1: Did authors try other noise metrics to evaluate the degree of noisy images? How is the performance like?__
>
> __A1:__ Yes, we also tried a naïve method to evaluate the degree of noisy images in the pre-experiments. This method uses the absolute value of the difference between the number of aspects in a sentence and the number of visual objects in an image to measure image noises. For simplicity, we call it noise_metric_diff. The corresponding performance on the JMASA task is below:
>
> Methods | TWITTER-15(Pre, Rec, F1) | TWITTER-17(Pre, Rec, F1) |
> | :--- | :----: | :----: |
> VLP-MABSA | 64.1 68.6 66.3 | 65.8 67.9 66.9 |
> VLP-MABSA + noise_metric_diff | 67.2 66.4 66.8 | 67.6 67.0 67.3 |
> VLP-MABSA + M2DF | 67.0 68.3 67.6 | 67.9 68.8 68.3 |
>
> The above results show noise_metric_diff brings some improvements, but it is still significantly inferior to our carefully designed framework M2DF.
>
> __Q2: Implementation details can be put into the appendix to leave some space for the analysis part. Case studies and error analysis are better to be put into the main content.__
>
> __A2:__ Thanks for your kind suggestion on structuring our paper. We will adjust the placement of implementation details, case studies, and error analysis to improve the overall presentation of our work in the revised version.
>
>
> __Q3: Authors should explain more on why we need curriculum learning in this work. Can we use other machine learning algorithms?__
>
> __A3:__ Thanks for your valuable feedback. As mentioned in the introduction, hard noise filtering using thresholds may lose some useful information. We use curriculum learning because it can achieve denoising without the need for data filtering. Specifically, it achieves denoising by adjusting the order of training data and reducing the exposure of noisy data during training. More theoretical analysis can be found in the paper "Why curriculum learning work in big/noisy data: A theoretical perspective"[3]. Up to now, we have not found other machine learning algorithms to achieve denoising without filtering the data.
>
> [3] Why curriculum learning work in big/noisy data: A theoretical perspective. Big Data & Information Analytics, 2016.
>
> __Q4: Readers might wonder whether other training schedules such as self-paced learning work.__
>
> __A4:__ Different from the two predefined task-specific noise metrics in M2DF, self-paced learning uses the training loss value as an automatic noise metric. To address your concern, we train the VLP-MABSA using self-paced learning on the JMASA task, and the results are as below:
>
> Methods | TWITTER-15(Pre, Rec, F1) | TWITTER-17(Pre, Rec, F1) |
> | :--- | :----: | :----: |
> VLP-MABSA | 64.1 68.6 66.3 | 65.8 67.9 66.9 |
> VLP-MABSA + self-paced learning | 65.8 67.1 66.5 | 65.9 68.5 67.2 |
> VLP-MABSA + M2DF | 67.0 68.3 67.6 | 67.9 68.8 68.3 |
>
> We can observe that the results of VLP-MABSA + self-paced learning are slightly higher than VLP-MABSA, but far inferior to VLP-MABSA + M2DF. This is reasonable because self-paced learning adjusts the learning schedule only based on training loss, and training loss is difficult to reflect whether the image contains noise in the MABSA task. In contrast, our designed task-specific noise metrics in M2DF help reduce the exposure to noisy data and achieve better performance.
>
> __Q5: More training schedules should be involved as baselines in the experiments to ensure fair comparisons since this work is a new training schedule.__
>
> __A5:__ Thanks for your precious suggestion. Apart from the above self-paced learning mentioned in A4, we also tried cross-entropy based training schedules [4] (abbr. CL_CE). The results on the JMASA task are as follows:
>
> Methods | TWITTER-15(Pre, Rec, F1) | TWITTER-17(Pre, Rec, F1) |
> | :--- | :----: | :----: |
> VLP-MABSA | 64.1 68.6 66.3 | 65.8 67.9 66.9 |
> VLP-MABSA + CL_CE | 66.3 66.9 66.6 | 66.5 68.0 67.2 |
> VLP-MABSA + M2DF | 67.0 68.3 67.6 | 67.9 68.8 68.3 |
>
> We can observe that the results of VLP-MABSA + CL_CE are inferior to VLP-MABSA + M2DF. The reasons are similar to self-paced learning, i.e., the value of the cross entropy is difficult to decide whether the image contains noise in the MABSA task. We will incorporate these results into our revision. Thanks for your comment again.
>
> [4] Uncertainty-aware curriculum learning for neural machine translation. ACL 2020.

---

### Official Review · Reviewer_ANSt · 2023-08-05

**Soundness:** 4

**Excitement:**

4: Strong: This paper deepens the understanding of some phenomenon or lowers the barriers to an existing research direction.

**Paper Topic And Main Contributions:**

Problem: There are many noisy images unrelated to the text in the dataset, which will have a negative impact on model learning.
Solution: The negative impact of noisy images can be reduced without filtering the data.
Contributions: They propose a Multi-grained Multi-curriculum Denoising Framework (M2DF), which can achieve denoising by adjusting the order of training data. They conduct extensive experiments on multiple datasets, and the results demonstrate the effectiveness of the proposed method.

**Reasons To Accept:**

1. The paper is well organized and easy to follow.
2. The proposed method is novel. They propose an effective approach to leverage all useful data.
3. The extensive results show the proposed method can be easily combined with existing methods and obtain better results.


**Reasons To Reject:**

Lack of comparison with other threhold-based data filtering methods

**Reproducibility:**

4: Could mostly reproduce the results, but there may be some variation because of sample variance or minor variations in their interpretation of the protocol or method.

**Reviewer Confidence:**

3: Pretty sure, but there's a chance I missed something. Although I have a good feel for this area in general, I did not carefully check the paper's details, e.g., the math, experimental design, or novelty.

---

> ### Author Rebuttal · Authors · 2023-08-28
>
> Thanks for your appreciation of our well-organized paper, novel method, and better results. We answer your concerns as follows:
>
> __Q1: Comparison with other threshold-based data filtering methods.__
>
> __A1:__ Thanks for your valuable comment. Following your suggestion, we added two threshold-based data filtering methods (RDS[1] and ITM[2]) as baselines and extended our framework to them. Among them, RDS is designed for information extraction tasks, and ITM is proposed for classification tasks, so we conducted experiments on JMASA and MASC tasks respectively. To be specific, we replaced the threshold-based data filtering module in RDS and ITM with our M2DF framework and named them as RDS(M2DF) and ITM(M2DF).
>
> For the JMASA task:
>
> Methods | TWITTER-15(Pre, Rec, F1) | TWITTER-17(Pre, Rec, F1) |
> | :--- | :----: | :----: |
> RDS | 60.8 61.7 61.2 | 61.8 62.9 62.3 |
> RDS(M2DF) | 61.2 63.0 62.1 | 62.4 63.6 63.0 |
>
> For the MASC task:
>
> Methods | TWITTER-15(Acc, F1) | TWITTER-17(Acc, F1) |
> | :--- | :----: | :----: |
> ITM | 78.3 74.2 | 72.6 72.0 |
> ITM(M2DF) | 78.9 75.0 | 73.2 73.0 |
>
> Compared to the baselines RDS and ITM, RDS(M2DF) and ITM(M2DF) show consistent improvements. These results again demonstrate that the denoising effect of M2DF is better than that of the threshold-based data filtering module. We will incorporate the above results into the revision. Thanks for your suggestion again.
>
> [1] Different Data, Different Modalities! Reinforced Data Splitting for Effective Multimodal Information Extraction from Social Media Posts. COLING 2022.
>
> [2] Targeted Multimodal Sentiment Classification based on Image-Target Matching. IJCAI 2022.

---

### Meta-Review · Area_Chair_T8Xw · 2023-09-26

**Recommendation:** 4

**Metareview:**

The reviewers found the paper to be strong/good across all metrics. All reviewers highlighted the novelty of the work. Reviewers were mixed on the quality of the writing. Two reviewers noted the paper lacks comparisons with other methods.

---

### Decision · Program_Chairs · 2023-10-07

**Decision:**

Accept-Main

**Comment:**

The reviewers found the paper to be strong/good across all metrics. All reviewers highlighted the novelty of the work. Reviewers were mixed on the quality of the writing. Two reviewers noted the paper lacks comparisons with other methods.